# Maternal environment alters dead pericarp biochemical properties of the desert annual plant *Anastatica hierochuntica* L.

**Janardan Khadka**[1☯], **Buzi Raviv**[1☯], **Bupur Swetha**[1], **Rohith Grandhi**[1], **Jeevan R. Singiri**[1], **Nurit Novoplansky**[1], **Yitzchak Gutterman**[1], **Ivan Galis**[2], **Zhenying Huang**[3], **Gideon Grafi**[1]*

**1** French Associates Institute for Agriculture and Biotechnology of Drylands, Jacob Blaustein Institutes for Desert Research, Ben-Gurion University of the Negev, Midreshet Ben Gurion, Israel, **2** Institute of Plant Science and Resources, Okayama University, Kurashiki, Okayama, Japan, **3** State Key Laboratory of Vegetation and Environmental Change, Institute of Botany, Chinese Academy of Sciences, Beijing, China

☯ These authors contributed equally to this work.
\* ggrafi@bgu.ac.il

**Data Availability Statement:** All data regarding this study are included in the manuscript and Supporting Information.

## Abstract

The dead organs enclosing embryos (DOEEs) emerge as central components of the dispersal unit (DU) capable for long-term storage of active proteins and other substances that affect seed performance and fate. We studied the effect of maternal environment (salt and salt+heat) on progeny DU (dry indehiscent fruit) focusing on pericarp properties of *Anastatica hierochuntica*. Stressed plants displayed increased seed abortion and low level and rate of germination. Hydrated pericarps released antimicrobial factors and allelopathic substances that inhibit germination of heterologous species. Proteome analysis of dead pericarps revealed hundreds of proteins, among them nucleases, chitinases and proteins involved in reactive oxygen species detoxification and cell wall modification. Salt treatment altered the composition and level of proteins stored in the pericarp. We observed changes in protein profile released from seeds of salt-treated plants with a notable increase in a small anti-fungal protein, defensin. The levels of phytohormones including IAA, ABA and salicylic acid were reduced in dead pericarps of stressed plants. The data presented here highlighted the predominant effects of maternal environment on progeny DUs of the desert plant *A. hierochuntica*, particularly on pericarp properties, which in turn might affect seed performance and fate, soil fertility and consequently plant biodiversity.

## Introduction

The basic unit of dispersal in higher plants is the seed, whereby the embryo is covered with maternally-derived dead organs including the seed coat (e.g., dehiscent dry fruit), fruit coat (pericarp, indehiscent dry fruits) and glumes, lemmas and paleas (spikelets and florets) in grasses. Although, the term dispersal unit emphasizes the capacity for dispersal in the habitat,

**Funding:** The work was supported by a Joint NSFC-ISF Research Grant to GG (grant no. 2456/18) and ZH (grant no. 31861143024). This work was partly supported by the Joint Usage/Research Center, Institute of Plant Science and Resources, Okayama University, Japan. The funders had no role in study design, data collection and analysis, decision to publish, or preparation of the manuscript.

**Competing interests:** The authors have declared that no competing interests exist.

this unit serves multiple functions including embryo protection from predation and from pathogen invasion, as well as from hazardous environmental condition [1].

The dead covering layers of the dispersal unit (e.g., seeds, fruits, florets, spikelets) may contribute to dormancy and/or inhibit germination due to barriers preventing water uptake or gaseous exchange, germination inhibitory substances [2–6] or due to mechanical barriers preventing embryo expansion [7,8]. In recent years, accumulating data suggest that the maternally-derived dead organs enclosing embryos (DOEEs) both in monocot and dicot plants function as a long-term storage for active hydrolytic enzymes and other substances that could contribute to seed persistence in the soil, germination and seedling establishment [9–12]. Several studies highlighted DOEEs as a storage for nutrients such as potassium, calcium and other cations that move into the seed during imbibition [9,13]. Other reports showed that the intact dispersal units improved the seedling establishment compared to naked seeds [9,14,15]. In a recent study, El-Keblawy et al. [16] examined the importance of husks (dead lemma and palea) and of the membranes surrounding *Brachypodium hybridum* caryopsis on germination and seedling growth. They found that the husks, particularly soaked husks significantly improved all examined parameters including final germination, germination rate index as well as seedling growth, concluding that it would be beneficial for farmers growing *B. hybridum* as a forage and cover crop to sow soaked husked grains. Yet the effect of husks on germination may be conflicting. Accordingly, earlier work on rice showed that the rice husks contain substances that play a significant role in inhibiting seed germination of various plant species including rice and lettuce [17]. On the other hand, Ueno and Miyoshi [18] reported that germination of intact rice seeds collected at 45 days after anthesis is significantly higher than those of dehusked seeds.

It has been well documented that environmental conditions to which mother plants exposed during vegetative stage or during seed production have a significant impact on the biological properties of progeny seeds [19–21]. Accordingly, seeds developed under drought conditions often generate higher seedling vigor compared to seeds of non-stressed plants [22,23]. Exposure of *Arabidopsis* plants to lower temperature (16°C) during the vegetative phase and during seed set caused a large increase in progeny seed dormancy [24]. Also, photoperiodical treatments of various plant species during seed maturation affected the properties of progeny seeds including seed coat color, structure, water permeability and germination capacity [25–27].

Climate change is anticipated to increase the frequency and severity of abiotic stresses including drought, salinity and temperature extremes, prompting us to investigate the effect of maternal growth conditions on DOEE properties. We hypothesized that exposure of mother plants to variable environmental conditions during vegetative phase and/or during seed production might have an impact on the composition and level of proteins and other substances accumulated within DOEEs and consequently affect progeny dispersal unit performance and fate. We selected the annual desert plant *Anastatica hierochuntica* (also known as the 'True Rose of Jericho', Brassicaceae) for this study, particularly because we wanted to examine the assumption that their adaptive capacity to variable desert environments may mitigate the effect of climate change on desert plant communities [28]. It is a Saharo-Arabian phytogeographical element that inhabits extremely xeric habitats near the Dead Sea and at Southern Negev in Israel. It has a raceme type of flowering and sessile inconspicuous flowers, which are essentially self-pollinated. The mature fruits carried on the dead skeleton are indehiscent and durable and store the seeds for many years. The seeds are released gradually from the fruit by a hygrochastic mechanism (which is dependent on the rainfall) and germinate rapidly on the soil surface [29]. The major aim of the present work was to evaluate the effect salt and salt+heat (S+H) on progeny dispersal unit properties with a focus on the pericarp. We employed various

methodologies including germination assays, microbial growth assays, in-gel assays, as well as proteomics and metabolomics to explore variation in pericarp properties induced by stress. Our data illuminated the elaborated function of the pericarp as a storage for multiple beneficial proteins and substances that can affect seed performance, and showed that maternal environment significantly affect the properties of the seeds and pericarps of a desert plant.

## Materials and methods

### Plant growth conditions and exposure to stress

*Anastatica hierochuntica* seeds were collected from a site in the Judean desert, near the dead sea, Israel, located outside of the Judean Desert nature reserve (31˚08'23.71"N 35˚21'47.87"E, at elevation of 347 m below the sea level; no permission was required for collecting seeds). Seeds were sown on trays containing standard gardening soil composed of peat and perlite (2:1 ratio) supplemented with 5g/L slow-release fertilizer, in a growth room (12/12 photoperiod; 22˚C±2). The 2-week-old seedlings were transplanted into pots (1.5 L) and grown on a similar gardening soil in a greenhouse (28˚C ±3) from Feb to May 2019. Plants (each pot) were irrigated with 100 ml water every 2 days. At the onset of flowering, 2/3 of the pots were treated with salt by irrigating each pot, every 2 days, with 100 ml of salty water starting with 25 mM NaCl. The salt concentration was gradually increased to 200 mM NaCl, by doubling the concentration every week. The final treatment (200 mM NaCl) was applied for 3 weeks. For salt + heat (S+H) treatment, a portion of the salt-treated plants (at 200 mM NaCl) were exposed to 37 ˚C for 2 hr daily for 1 week. The rationale behind this heat treatment is based on meteorological data showing that during the growth season plants may experience high temperatures for several days [30]. Heat treatment was confirmed by analysis of HSP proteins by immunoblotting using anti-HSP70 (AS08 371, Agrisera AB, Vannas, Sweden) and anti-HSP17.6 (AS07 254, Agrisera AB, Vannas, Sweden).

After three weeks of 200 mM salt treatment, all plants were irrigated with water for 2 weeks and left to dry. Dried skeletons carrying the indehiscent fruits were collected and store at room temperature until used. Seeds of *Sinapis alba* were collected from natural populations growing near agricultural fields at northern Negev (Mabuiim; 31˚26'42"N 34˚39'37"E). *Brassica juncea* seeds were purchased from the local market.

### Extraction of plant materials and in gel assays

Pericarps collected from *Anastatica* dead skeleton were ground into fine particles and extracted (100 mg) in 0.5 ml of water or phosphate buffered saline (PBS) by incubation overnight at 4˚C with gentle shaking. Samples were centrifuged at 4˚C at maximum speed (14,000 rpm) for 10 min and the cleared supernatant was collected and sterilized by passing through 0.22 μm PVDF filter unit (Millex -GV, Merck Millipore, Tullagreen, Carrigtwohill, IRL). Extracts were used in germination assays, proteomic analysis, bacterial growth as well as for enzymatic activity using in-gel assays. In-gel nuclease, chitinase and protease assays were performed essentially as described [12].

### Germination assays

Germination of *A. hierochuntica* seeds were performed in 4 replicates each contains 32 seeds in a Petri dish on a blot paper supplemented with water in a growth room at 22˚C in the dark. Because *Anastatica* seeds are known to germinate rapidly following hydration [29], we inspected germination after 10, 14, 24, 36, 48 and 60 h, and calculated the rate and the final percentage of germination. The effect of extracts obtained from *A. hierochuntica* pericarps and

seeds on germination of heterologous seeds (*Sinapis alba* and *Brassica juncea*) was performed in three replicates (20 seeds each) in a Petri dish on a blot paper supplemented with water or with the abovementioned extracts. Germination was performed in the dark at 22˚C, inspected daily and photographed.

## Bacterial growth assay

The assay was performed essentially as described [31]. Briefly, *Escherichia coli* (ATCC 10978) were grown overnight on LB medium at 37˚C, the culture was diluted, transferred to 25% LB broth and grown at 37 ˚C to 0.03–0.05 optical density (OD595; Epoch, Biotek, Winooski, VT, USA). To a 150 μL aliquot of the culture 50 μL of LB (control 1) PBS (control 2), 50 μl of PBS + 25% Hoagland solution, kanamycin (final concentration 50 μg/ml), caryopsis extracts (50 μl) or with 50 μl filtered (through 0.2 μm) husk extract (three replicates per treatment) in a flat-bottom 96-well microtiter plate. Plates were incubated in the dark using a spectrophotometer (Synergy 4, Biotek, USA) and reads (OD595) were taken in intervals of 30 min in a course of 12 h. The average OD for each blank replicate at a given time point was subtracted from the OD of each replicate treatment at the corresponding time point and standard errors were calculated for each treatment at every time point.

## Proteome analysis

For proteome analysis, intact seeds (32 seeds) and 10 mg of ground pericarps were placed in 2 ml tube with 100 μl PBS and incubated at 4 ˚C for 1 h with gentle rotation, then centrifuged at 4˚C at maximum speed (14,000 rpm) for 10 min. 50 μl of supernatants were collected and stored at -20˚C until used for comparative, quantitative proteome analysis.

Proteome analysis of proteins released from *Anastatica* seeds (4 replicates) and pericarps (3 replicates) was performed by the proteomic services of The Smoler Protein Research Center at the Technion, Israel. Note, each replicate contains randomly selected seeds or pericarps from a pool of 14–18 plants. Proteins were digested with trypsin followed by separation and mass measurement on LC-MS/MS on LTQ-Orbitrap Protein identification and quantification were done using MaxQuant, using *Arabidopsis thaliana* proteins from UniProt as a reference. The file proteinGroups.txt (protein-level data) was subjected for further analysis. We started from protein-level LFQ-normalized intensities and used our in-house R script, with some more help from Excel and Partek Genomics Suite. The analysis included: Quality assessment of the raw and LFQ-normalized intensities, filtering out proteins marked as contaminant, reverse and only identified by site. This was followed by Log2 transformation of the LFQ intensities.

We further filtered proteins so that only proteins having at least 2 non-zero replicates in at least one of the treatment groups were retained. Present/absent analysis: identifying proteins which have zero values in all replicates of one of the treatment groups, and at least 2 non-zero values in another treatment group. This analysis assumes that 0 values in all replicates means that the protein is actually absent in that treatment group. Imputation of zero intensities by random numbers from a normal distribution, with width = 0.2 and downshift = 1.6. To avoid relying too much on artificial numbers, the entire dataset was submitted to imputation 10 times. In each repeat, the imputation produced different random numbers. Each of the 10 imputed dataset was submitted to hypothesis testing for differential protein expression using Limma. The statistical model tested the contrast between salt and control-treated plants. A protein was considered differentially present (DP) in a contrast if it passed the following cut-offs in at least 8 of the 10 imputed datasets. Unadjusted p-value < 0.01 Linear fold change < -1.3 or > 1.3 (where minus sign indicates down-regulation). Proteins having non-zero LFQ

values in at least 3 samples (out of 6, regardless of treatment group) were submitted to GO annotation and functional categorization using Gene Ontology at TAIR [32].

**Resources.**  All the identified peptides were filtered with high confidence, top rank, mass accuracy, and a minimum of 2 peptides. High confidence peptides were passed the 1% FDR threshold (FDR = false discovery rate, is the estimated fraction of false positives in a list of peptides). Semi-quantitation was done by calculating the peak area of each peptide. The area of the protein was calculated from the average of the three most intense peptides from each protein.

## Phytohormone analysis

Plant tissues were chopped into small pieces and ground into fine powder in liquid nitrogen. Phytohormone extraction was done essentially as described in Fukumoto et al. [33]. Briefly, approximately 50 mg of each sample was extracted with 1 ml ethyl acetate supplemented with deuterium labeled internal standards (IS) (10 ng D3-JA, 5 ng D3-JA-Ile, 10 ng D6-ABA and 20 ng D4-SA). Zirconia beads were added to the tube and samples were homogenized for 2 min in FastPrep instrument, centrifuged at 13,200 g for 15 min at 4 ˚C and supernatant was collected. Second extraction was done with 0.5 ml ethyl acetate as above and supernatants of both extractions were pooled. Solvent was evaporated by vacuum centrifugation and extract was dissolved in 300 μl of 70% methanol. The mixture was diluted by adding 1.7 ml of 84 mM ammonium acetate buffer (pH 4.8), and purification was done by applying the samples to preconditioned Bond Elut C18 columns. Phytohormones was eluted with 0.8 ml of 85% methanol. The samples were dried by vacuum centrifugation and dissolved in 0.1 ml of 70% methanol.

**Extraction method 2.**  Tissues were processed as above. Approximately 50 mg of each sample was taken and 4 ml of extraction solution (80% acetonitrile, 1% acetic acid) containing internal standard (IS) mixture (D2-GA1, D2-GA4, D5-tZ, D6-iP, D2-JA, D5-JA, $^{13}$C6-JA-Ile, D2-JA-Ile, D2-IAA, $^{13}$C6-IAA, D6-ABA, D4-SA) was added and samples were incubated for 1 h at 4 ˚C and centrifuged at 3000 g for 10 min at 4 ˚C. Second extraction was done in 4 ml of extraction solution without IS and both extracts were combined. Then acetonitrile was evaporated by storing samples at -30 ˚C, concentrated in vacuum evaporator for 100 min at 50 ˚C and 1% acetic acid was added to make a volume of 1.5 ml.

Solid phase extraction and LC-MS analysis were carried out essentially as described in [34] Gupta et al. (2018). Briefly, the extracted sample was applied into pre-equilibrated Oasis HLB cartridge (Waters Corporation, USA), washed with 1% acetic acid and phytohormones were eluted with 80% acetonitrile + 1% acetic acid. The acetonitrile was evaporated from the eluate using centrifugal vacuum evaporator and loaded onto pre-equilibrated Oasis MCX cartridge (Waters Corporation, USA). After washing with 1% acetic acid, the acidic fraction was eluted with 80% acetonitrile + 1% acetic acid. A fraction of acidic fraction was dried and reconstituted in 1% acetic acid for analysis of Salicylic Acid. The MCX cartridge was washed with 2% ammonia, and the basic fraction was eluted with 50% acetonitrile + 5% ammonia. The basic fraction was dried and reconstituted in 1% acetic acid for analysis of cytokinins. The remaining acidic fraction was processed for evaporation of acetonitrile and loaded onto pre- equilibrated Oasis WAX cartridge (Waters Corporation, USA). The WAX cartridge was washed with 1% acetic acid followed by 50% acetonitrile, and phytohormones were eluted with 80% acetonitrile + 1% acetic acid. Final SAWAX elution was done using 3% formic acid + acetonitrile. The eluates were dried and reconstituted in 1% acetic acid and subjected to analysis of ABA, IAA, GA4, JA, and JA-Ile.

**Measurement.**  The phytohormone quantification was performed in triple quadrupole LC-MS/MS system (Agilent Technologies, USA). Peak area of each endogenous phytohormones

and internal standards was calculated on MassHunter Qualitative Analysis software (Agilent). Phytohormone concentration was calculated by comparing peak areas of endogenous phytohormone to that of internal standards and expressed in per g dry-wet of sample.

## Metabolite analysis

Extraction and quantification of primary metabolites were performed using GC-MS method essentially as described [35]. Briefly, plant tissues were ground in liquid nitrogen and the samples (40 mg) were extracted in 1.4 ml of 100% methanol with ribitol (12 μg) supplemented as an internal standard. The samples were homogenized for 2 min, incubated at 70 ˚C with shaking for 10 min. The samples were centrifuged at 4 ˚C for 10 min at 12,000 g. The supernatant was taken and vigorously mixed with 0.7 ml chloroform and 1.5 ml water. The phases were separated by centrifugation at 2,000 g for 15 min and 150 μL of the upper polar phase was sampled and dried in vacuum concentrator at RT. The dried samples were sequentially derivatized with methoxyamine hydrochloride and N-methyl-N-trimethylsilyl-trifluoroacetamide. The derivatized samples were diluted (1/10) with dichloromethane and metabolites were analyzed with an Agilent 7890A-GC/240-MS instrument (Agilent Technology, USA). One microliter of each diluted sample was injected in split mode (1:10) into the injector port of the GC instrument held at 230 ˚C via an auto-sampler. Carrier gas (He) flow rate was set to 1 ml/min. Chromatography was performed on a HP-5MS capillary column (5% phenyl methyl silox, 30 m × 250 μm × 0.25 μm) (Agilent Technologies, USA). The GC oven temperature was programmed at 60 ˚C for 3 min, followed by a 5 ˚C/min ramp to 300 ˚C and a final 5 min heating at 300 ˚C before returning to initial conditions. Mass spectra data were collected in full scan mode in mass range $m/z$ 40–750. Metabolites were identified by comparing their fragmentation patterns with those of Mass Spectral Library (National Institute of Standards and Technology, USA). Co-injection with authentic standards was performed to confirm tentative identifications. Quantification of metabolite concentration was done based on standard curves generated for each target compound and internal standards.

## Statistical analysis

Statistical analysis (unpaired t test) was performed using the GraphPad QuickCalcs Web site: https://www.graphpad.com/quickcalcs/ttest1/?Format=C (accessed November 2019) or using the Microsoft Excel platform. All assays were repeated at least three times and representative results are shown.

## Results

### Effects of stress on germination of *Anastatica hierochuntica* seeds

To assess the effect of stress on substances accumulated in dead organs enclosing embryos (DOEEs) such as pericarps, we subjected the desert annual plant *A. hierochuntica* in the course of flowering and seed maturation to stress conditions commonly prevail in its ecosystem, namely, salt and combination of salt and heat (S+H); note heat stress was applied in accordance with its occasional occurrence for a few days during the growth season. No differences in average number of pods and seeds (normal and aborted) were found between control and stress-treated plants (Fig 1A and 1B). Yet, seed abortion (as determined by shrunken seed morphology) was higher in stress-treated plants showing 2.5, 8.3 and 12.9% of abortion in control, salt, and S+H-treated plants, respectively (Fig 1C). Furthermore, germination experiments revealed a significant reduction in germination of seeds derived from stress-treated plants. Accordingly, 93% of seeds from control plants were germinated within 24 h, while only

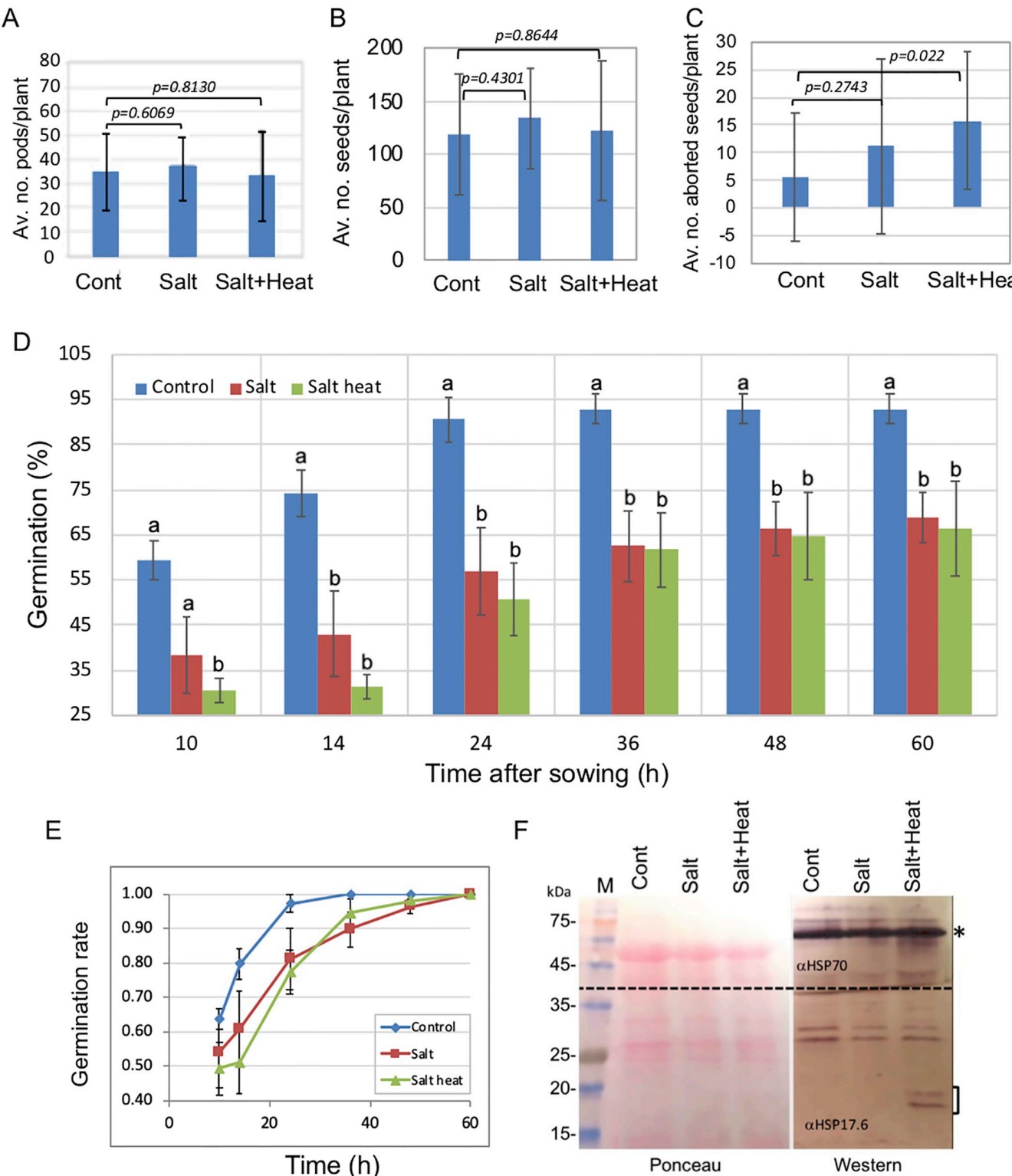

**Fig 1. Increased seed abortion and reduced seed germination following exposure of *A. hierochuntica* mother plants to stress conditions.** Average number of pods (A) and seeds (B) and percentage of seed abortion (C) following exposure of mother plants to stress conditions. Note, the average number of seeds (B) includes aborted seeds. n = 14, 16 and 18 for control, salt and salt+heat, respectively. Vertical bars represent the standard deviation (SD). Reduction in percentage of germination (D) and germination rate (E) of seeds derived from plants exposed to stress. Each bar value represents the mean ± SD of quadruplicate experiments (n = 32). Different letters at each time point indicate statistically significant differences between treatments (p < 0.05; Student's

unpaired t test). (F) Plants subjected to heat stress (2h 37° C/7 days) expressed small HSP17.6. Left panel is the ponceau staining of the membrane and right panel is the immunoblotting (Western). Note the membrane was cut into two (broken line), the upper containing proteins above 35 kDa was probed with anti HSP70 (αHSP70, marked with asterisk) and the lower part with anti HSP17.6 (αHSP17.6, marked with a bracket).

about 60% of seeds derived from salt and S+H-treated plants were germinated and their rate of germination slowed down two-fold, reaching a peak within 48 h (Fig 1D and 1E). We confirm the response to heat stress by immunoblotting using anti HSP70 and anti HSP17.6. Accordingly, only plants subjected to daily heat stress for 2 h in a course of 7 days expressed the small HSP17.6 (Fig 1F), which is known to be upregulated following exposure of plant to certain stress conditions [36].

## Dead pericarps release allelopathic substances affecting seed germination

We examined the effect of substances released from *A. hierochuntica* seeds on seed germination of *A. hierochuntica*, *Brassica juncea* and *Sinapis alba*. To this end, seeds were germinated on a blot paper supplemented with water or substances released from seeds (Ah seed-R) and inspected after 48h. Results showed that the effect of substances released from seeds is species specific (S1 Fig in S1 File). Accordingly, we found that substances released from Ah seeds did not affect germination of *A. hierochuntica* and *B. juncea* but seed germination of *S. alba*. Germination of *S. alba* seeds was recovered after extensive washing with water and incubation in water for another 48 h (S1 Fig in S1 File). Next, we examined the effect of substances released from *Anastatica* pericarps obtained from untreated plants or plants treated with salt and Salt+heat (S+H) on seed germination of *B. juncea* (Fig 2A) and *S. alba* (Fig 2B) in comparison to water. While all pericarp extracts had a strong inhibitory effect on germination of *S. alba*, these extracts differentially affected germination of *B. juncea* seeds. Accordingly, pericarp extracts from control plants had almost no effect on seed germination of *B. juncea*. However, pericarp extracts from salt-treated plant significantly reduced germination at all time points examined, while pericarp extracts from S+H-treated plants strongly inhibited germination of *B. juncea*. Seed germination of *S. alba* incubated in all pericarp extracts was strongly inhibited showing no germination after 72 h (Fig 2B) compare to seed germination in water.

## Dead pericarps of *A. hierochuntica* release hundreds of proteins upon hydration: The effect of maternal growth conditions

Proteins released from pericarps of control and stress-treated *Anastatica* plants were subjected to proteome profiling using LC-MS/MS on LTQ-Orbitrap followed by identification by Discoverer software against the *Arabidopsis thaliana* proteome from the Uniprot and a decoy database (S1 Table in S2 File). We used the raw intensity data to perform a principle component analysis (PCA) to examine the effect of exposure of mother plants to stress conditions, on accumulation of proteins in the pericarps (Fig 3A). Results showed that principal component 1 (PC1) explaining 36.5% of the variance separated the control from salt-treated plants, demonstrating the notable effect of stress conditions on proteins accumulated within the dead pericarps of *A. hierochuntica*. Implementing the cutoffs (No. peptides >1, except for proteins whose molecular weight is below 10 kDa), we identified 502 proteins released from dead pericarps (S2 Table in S2 File). Functional categorization for biological process highlighted that among the 447 proteins recognized in this category, 166 proteins are involved in response to stress (S2A Fig in S1 File). Molecular function analysis highlighted several groups of proteins including hydrolases (135 proteins) and RNA and DNA binding proteins (49 proteins) and

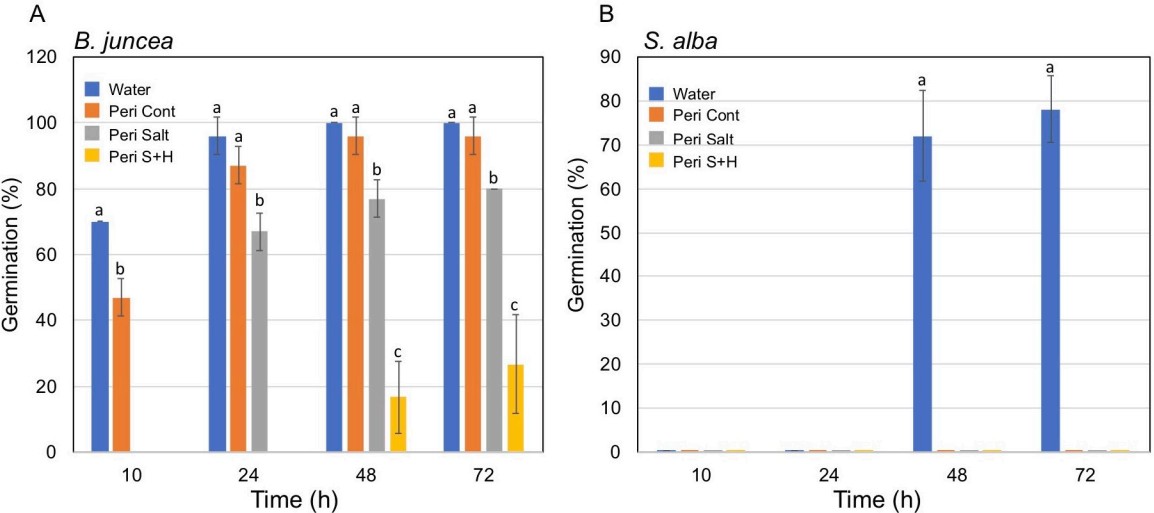

**Fig 2. Dead pericarps of *A. hierochuntica* contain allelopathic substances differentially inhibiting germination of heterologous seeds.** Seeds of *Brassica juncea* (A) and *Sinapis alba* (B) were germinated for 72 h in water or in pericarp extracts derived from mother plants grown under control (Peri Cont), salt (200 mM, Peri salt) or Salt+heat (Peri S+H) conditions. Germination was recorded after 10, 24, 48 and 72 h. Vertical bars represent the standard deviation of triplicate experiments (n = 20). Different letters at each time point indicate statistically significant differences between treatments (p < 0.05; Student's unpaired t test).

protein binding (129 proteins) (S2B Fig in S1 File). Multiple hydrolytic enzymes listed in the proteome data (S3 Table in S2 File) include endonuclease 1, a homolog of the *Arabidopsis* BFN1, BIFUNCTIONAL NUCLEASE I (ENDO1, At1g11190) involved in leaf senescence and cell death [37]. Also, we identified class V chitinase and the basic endochitinase B (homolog of

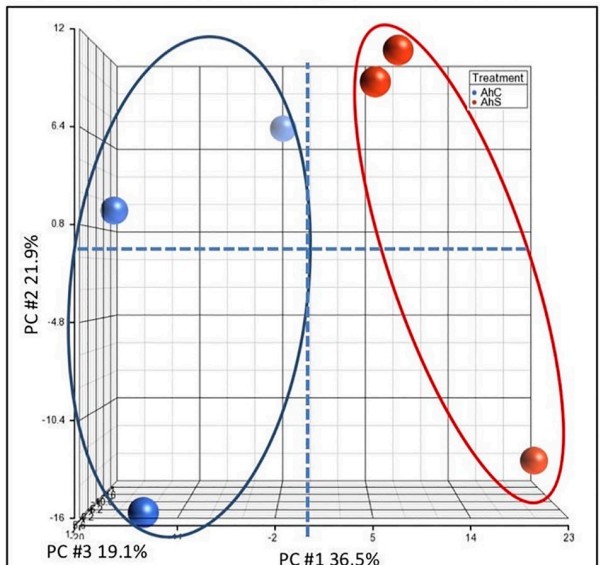

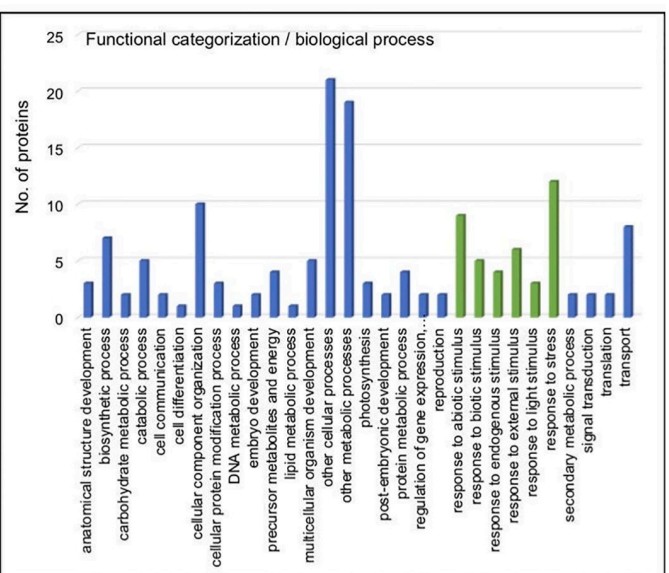

**Fig 3. Principal Component Analysis (PCA) score plots comparing the proteome profiles between mother plants of *A. hierochuntica* exposed to stress conditions with control, untreated plants.** **A**. PCA demonstrating the proteins released from pericarps of control (AhC, blue dots) and salt-treated plants (AhS, red dots). **B**. Biological process categorization of differentially present (DP) proteins in dead pericarps derived from control and salt-treated plants. Categories related to response to various stimuli are highlighted green.

PR-3, encoded by At3g12500), involved in chitin catabolism and in response to biotic and abiotic stresses [38], as well as multiple types of proteases including serine protease subtilisin-like protease SBT1.6, aspartic protease related to the Arabidopsis At5g07030 gene product as well as cathepsin B-like protease, a caspase protein involved in stress induced cell death [39]. Other protein groups include heat shock proteins (HSP17.6; HSP90-2), ROS detoxifying enzymes and cell wall modification enzymes (S3 Table in S2 File).

We identified 38 differentially present (DP) proteins released from pericarps derived from control and salt-treated plants (Table 1; S4 Table in S2 File). Among these DP proteins, 15 were downregulated and 23 were upregulated. Functional categorization for cellular process showed (Fig 3B) that 31% of the DP proteins are related to response to biotic and abiotic stresses including polygalacturonase inhibitor 2, an important factor for plant resistance to phytopathogenic fungi [40], PLAT domain-containing protein 1, a positive regulator of abiotic stress tolerance [41] as well as annexin, a calcium binding protein upregulated in multiple stresses [42].

We also performed proteome analysis of the proteins released from *A. hierochuntica* seeds derived from parental plants exposed to stress conditions. Accordingly, 4 replicates from control, salt and S+H were analyzed by LC-MS/MS on LTQ-Orbitrap followed by identification by Discoverer software against the *Arabidopsis thaliana* and the Brassicaceae proteome from the Uniprot and a decoy database (S5 Table in S2 File). We used raw intensities with 2 or more peptides to perform a PCA to compare the responses of *A. hierochuntica* to stress conditions (S3A Fig in S1 File). Results showed that while in control plants each independent replicate clustered together, the distribution of the Salt and S+H samples is highly diverse. Principal Component 1 (PC1) explaining 49.9% of the variance separates the control from stress-treated plants. Thus, similarly to the results obtained from pericarps, the seed PCA demonstrates the significant effect of maternal stress conditions on proteins released from *A. hierochuntica* progeny seeds.

We identified 484 proteins released from seeds following implementation of the cutoffs (No. peptides >1, except for proteins whose molecular weight is below 10 kDa) [S6 Table in S2 File]. Among these proteins, several protein groups are highlighted (S7 Table in S2 File). These include nucleases (e.g., endonuclease 1, ribonuclease Tudor 1/2), chitinases, proteases, ROS detoxifying enzymes, cell wall modification enzymes and multiple heat shock proteins. We identified 21 DP proteins that were released from seeds derived from control and stress-treated plants (S8 Table in S2 File). Among these DP proteins, 20 proteins were downregulated many of which are ribosomal proteins and some are related to stress response (S3B Fig in S1 File). Interestingly, only 1 protein was significantly upregulated in salt and S+H treated plants, that is, defensin-like protein 195, which belongs to a small, cysteine-rich protein family that possesses antifungal activity [43,44].

## Maternal growth conditions affect activity of hydrolases released from dead pericarps upon hydration

As mentioned above, proteome data highlighted certain protein groups released from seeds and dead pericarps including nucleases and chitinases (S3 and S7 Tables in S2 File). We therefore performed in gel assays to examine the activities of hydrolases released from seeds and dead pericarps derived from control and stress-treated plants. Results showed clear reduction in nuclease activity in pericarps derived from stress-treated plants (Fig 4A), while no significant change was evident in chitinase activity (Fig 4B). We observed a notable increase in protease activity in pericarps derived from plants grown under S+H stress conditions.

**Table 1. Differentially Present (DP) proteins in pericarps of *A. hierochuntica* grown under control and salt conditions that pass the cutoff of unadjusted p-value < 0.05 and Fold Change (FC) > 2.**

| Protein ID | Protein.name | Arabidosis ID | pvalue S/C | Fold change | Category |
|---|---|---|---|---|---|
| F4INW2 | Winged-helix DNA-binding protein, Histone H1.2 | AT2G30620 | 0.00746 | 26.1 | Gene expression |
| B3H4B6 | 40S ribosomal protein S25 | AT4G39200 | 9.66E-05 | 24.2 | Translation |
| O82230 | Nucleoid-associated protein | AT2G24020 | 0.00695 | 19.9 | transport |
| Q9CAX7 | Small nuclear ribonucleoprotein G | AT3G11500 | 0.00243 | 15.6 | response to stress |
| Q9ZPY5 | Uncharacterized protein | AT2G46540 | 1.86E-05 | 14.5 | mitochondria |
| Q9SUU5 | Cytochrome b-c1 complex subunit 7–1 | AT4G32470 | 0.000238 | 13.6 | aerobic respiration |
| Q9CA23 | Ubiquitin-fold modifier 1 | AT1G77710 | 0.000737 | 10.9 | protein ufmylation |
| A8MS03 | Ribosomal protein S6 | AT4G31700 | 0.000867 | 9.19 | translation |
| Q9ZU25 | Probable mitochondrial-processing peptidase subunit alpha-1, insulinase | AT1G51980 | 0.000993 | 8.49 | proteolysis |
| A8MS83 | Ribosomal protein L23AB | AT3G55280 | 0.00376 | 8.08 | respons to stress |
| Q945K7 | Isocitrate dehydrogenase [NAD] catalytic subunit 5, mitochondrial | AT5G03290 | 0.00374 | 7.73 | TCA cycle |
| F4K3R8 | Voltage dependent anion channel 2 | AT5G67500 | 0.00285 | 7 | transmembrane transport |
| F4III4 | MALE GAMETOPHYTE DEFECTIVE 1 | AT2G21870 | 0.00448 | 6.34 | pollen development |
| Q9SR73 | 40S ribosomal protein S28-1 | AT3G10090 | 0.00626 | 6.34 | translation |
| F4JTQ0 | Vacuolar proton pump subunit B | AT4G38510 | 0.00444 | 5.57 | Ion transport |
| A0A1P8AXC1 | FtsH extracellular protease family | AT2G30950 | 0.0023 | 5.52 | proteolysis |
| O80800 | Acyl carrier protein 2, mitochondrial | AT1G65290 | 0.00262 | 4.91 | cellular proc. |
| Q9XGM1 | V-type proton ATPase subunit D | AT3G58730 | 0.00253 | 4.5 | transport |
| Q9MA79 | Fructose-1,6-bisphosphatase, cytosolic | AT1G43670 | 0.00136 | 4.45 | response to chemical |
| F4KGH1 | Annexin D2. | AT5G65020 | 0.0043 | 4.3 | response to stress |
| O65660 | PLAT domain-containing protein 1 | AT4G39730 | 0.00456 | 4.26 | response to stress |
| Q9M5J8 | Polygalacturonase inhibitor 2. | AT5G06870 | 0.00755 | 3.99 | signal trunsd. |
| Q39258 | V-type proton ATPase subunit E1 | AT4G11150 | 0.00744 | 3.93 | response to stress |
| Q94EG6 | Uncharacterized protein At5g02240 | AT5G02240 | 0.0065 | -3.08 | Response to chemical |
| O50008 | 5-methyltetrahydropteroyltriglutamate—homocysteine methyltransferase 1 | AT5G17920 | 0.00393 | -3.53 | response to chemical |
| Q9ZRW8 | Glutathione S-transferase U19 | AT1G78380 | 0.00641 | -3.81 | response to stress |
| Q9FRL8 | Glutathione S-transferase DHAR2 | AT1G75270 | 0.00137 | -4.42 | response to stress |
| Q0WW26 | Coatomer subunit gamma | AT4G34450 | 0.000575 | -5.29 | protein transport |
| Q9M9K1 | Probable 2,3-bisphosphoglycerate-independent phosphoglycerate mutase 2 | AT3G08590 | 0.000679 | -5.97 | response to chemical |
| F4I576 | Monodehydroascorbate reductase 6 | AT1G63940 | 0.00257 | -6.21 | response to stress |
| P43296 | Cysteine protease RD19A | AT4G39090 | 0.00582 | -6.3 | response to stress |
| O64743 | Berberine bridge enzyme-like 15 | AT2G34790 | 0.00614 | -6.4 | redox process |
| A0A1P8B956 | Amine oxidase | AT4G12290 | 0.00201 | -6.67 | redox process |
| F4JTH0 | Aspartate aminotransferase | AT4G31990 | 0.00313 | -7.25 | response to stress |
| Q9SVG4 | Berberine bridge enzyme-like 19 | AT4G20830 | 0.00296 | -9.26 | response to stress |
| Q8W493 | Ferredoxin—NADP reductase, leaf isozyme 2 | AT1G20020 | 0.000515 | -11.8 | response to stress |
| Q9XI10 | DPP6 N-terminal domain-like protein | AT1G21680 | 0.00172 | -16.4 | putative translation |
| P13114 | Chalcone synthase | AT5G13930 | 7.32E-06 | -29.8 | response to stress |

The related *Arabidopsis* gene ID is given. FC is fold change between pericarp of salt versus control; (-) indicates downregulation. Category was dissected based on gene ontology (GO) annotations of the *Arabidopsis* genes using GO at TAIR.

## Changes in accumulation of phytohormones and primary metabolites in dead pericarps of *A. hierochuntica* following exposure to stress

Pericarps derived from *A. hierochuntica* untreated and salt-treated plants were subjected to phytohormone analysis using liquid chromatography–mass spectrometry (LCMS). This

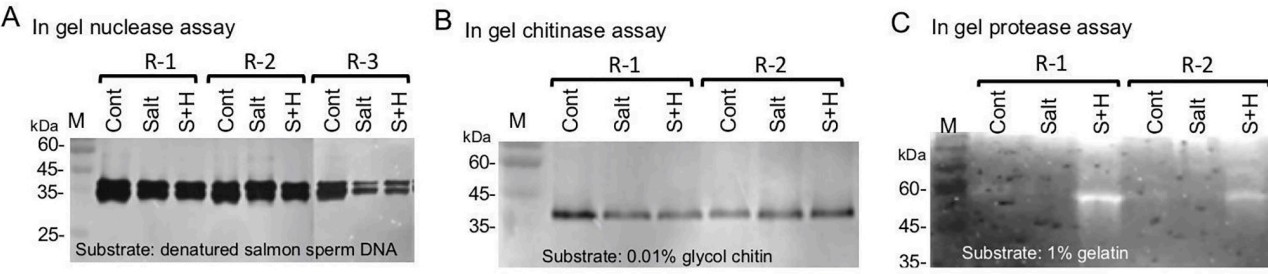

**Fig 4. Analysis of hydrolase activities in pericarps derived from stress-treated plants by in gel assays.** In gel nucleas assay (A), chitinase assay (B) and protease assay (C) were performed on proteins released from pericarps derived from control (Cont) plants or plants treated with salt or with a combination of salt and heat (S+H). Pericarps from two separate experiments R-1 and R-2 were analyzed. R-3 in A refers to experiments were all replicas were combined and analyzed with reduced extract concentration. Substrates used for in-gel assays are indicated at the bottom of each panel. M, protein molecular weight markers.

analysis revealed that the dead pericarps store multiple phytohormones including Abscisic acid (ABA), indole acetic acid (IAA), salicylic acid (SA), jasmonic acid (JA) and cytokinins [trans-zeatin (tZ) and dihydrozeatin (DHZ)]. We observe changes in phytohormone levels including significant reduction in ABA and IAA in pericarps of salt-treated plants (Fig 5).

## Phytohormones recovered from dead pericarps of *A. hirochuntica*

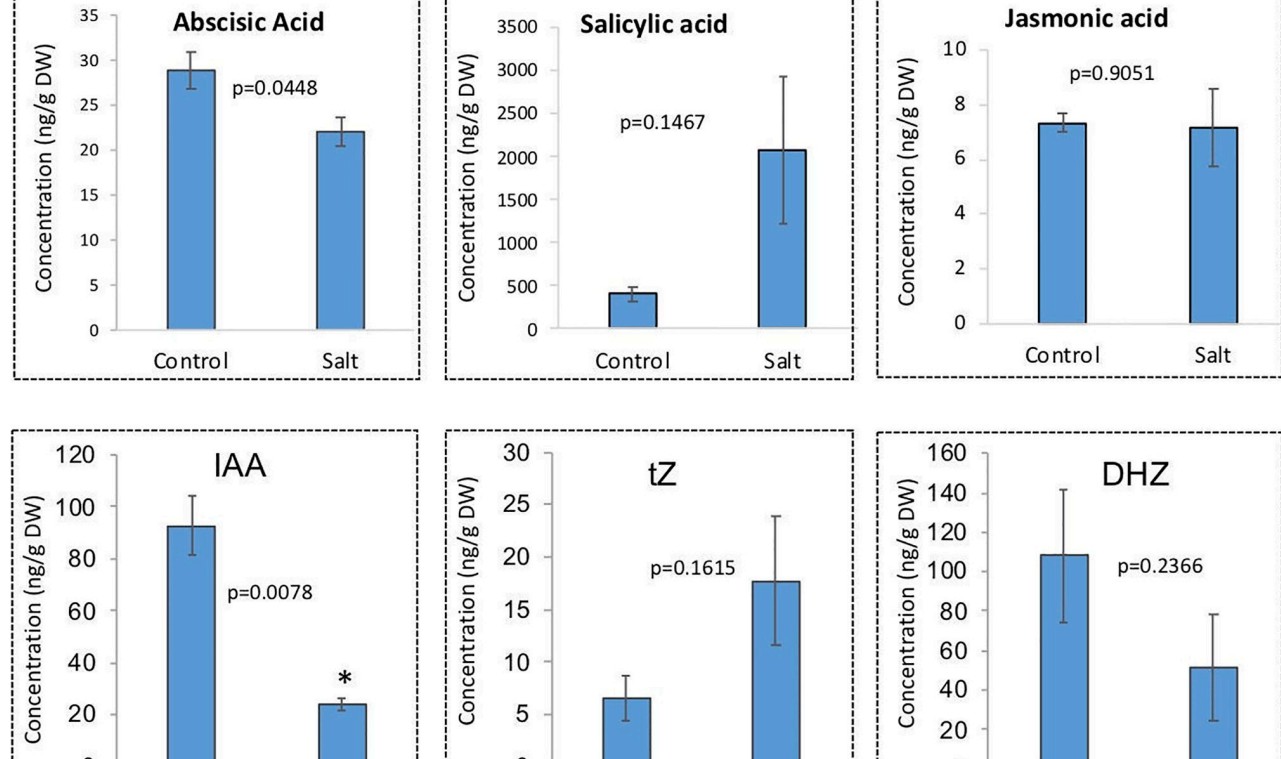

**Fig 5. Phytohormones recovered from dead pericarps obtained from control and salt-treated plants of *A. hierochuntica*.** IAA, indole acetic acid; tZ, trans zeatin; DHZ, dihydrozeatin. p-values below 0.05 are considered statistically significant. Bars represent the standard deviation.

Although not statistically significant, we observed increase in levels of SA and tZ and decrease in DHZ, while JA level was not changed. Further analysis of primary metabolites showed that the accumulation of citrate, succinic acid, malic acid and fumaric acid, all are intermediates of the tricarboxylic acid (TCA) cycle, was significantly reduced in pericarps derived from salt-treated plants (Fig 6).

# The citric acid cycle / Krebs cycle

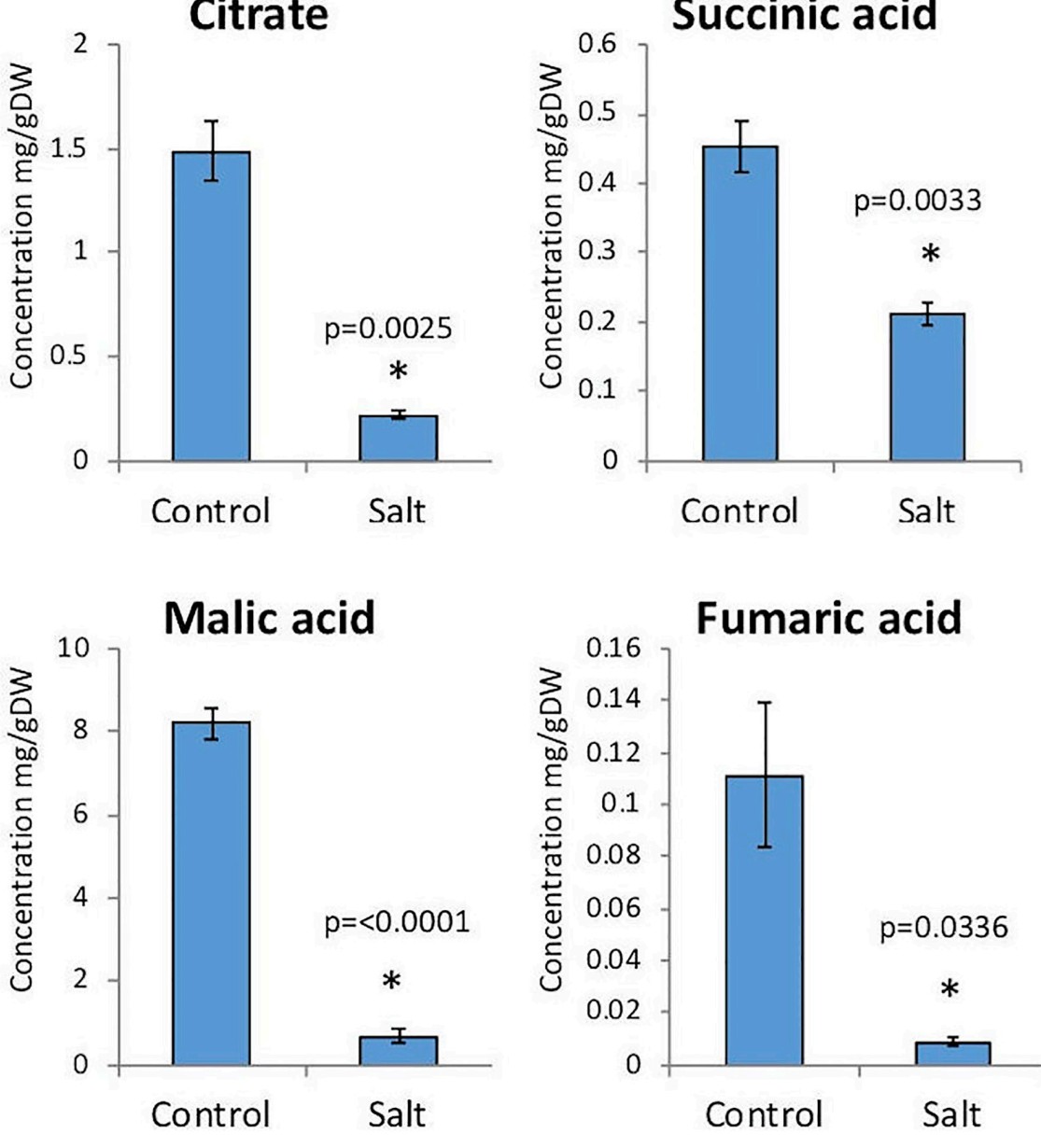

**Fig 6. Tricarboxylic Acid (TCA) cycle intermediates in pericarps derived from control and salt-treated plants of *A. hierochuntica*.** p-values below 0.05 are considered statistically significant. Bars represent the standard deviation. Note, all examined intermediates of the TCA cycle were significantly reduced in pericarps of salt-treated plants.

GCMS analysis of metabolites showed the accumulation of free amino acids in pericarps of *A. hierochuntica*, but their levels in the pericarps were not changed significantly following exposure of mother plants to salt stress (S4 Fig in S1 File).

The metabolic analysis also revealed the presence of multiple sugars including fructose, glucose, galactose and sucrose in pericarps with a notable elevation in their levels in pericarps derived from salt-treated plants (S5 Fig in S1 File).

### Effect of substances released from *Anastatica* seeds and dead pericarps on bacterial growth

Microorganisms have a range of inhibitory and stimulatory effects on early seed germination and seedling growth [45]. We wanted to examine for the presence of microbial growth controlling activity in substances released from seeds and pericarps of *Anastatica* plants grown under control and salt conditions. We used the gram-negative strain *E. coli* for our experiment. Bacteria were grown in a flat-bottom 96-well microtiter plate in LB medium supplemented with PBS, ampicillin (50 µg/ml), pericarp extracts or seed secretions. Plates were incubated in the dark using a Synergy 4 spectrophotometer (Biotek, USA) and reads ($OD_{595}$) were taken at 30 min intervals in a course of 12 h. Generally, growth of *E. coli* was inhibited by pericarp and seeds extracts compared to control PBS addition with a slight reduction in severity in extracts obtained from salt-treated plants (Fig 7). Similarly to ampicillin, pericarp extracts appears to possess strong inhibitory effect on bacterial growth, which was higher than the inhibitory effect of seed secretions.

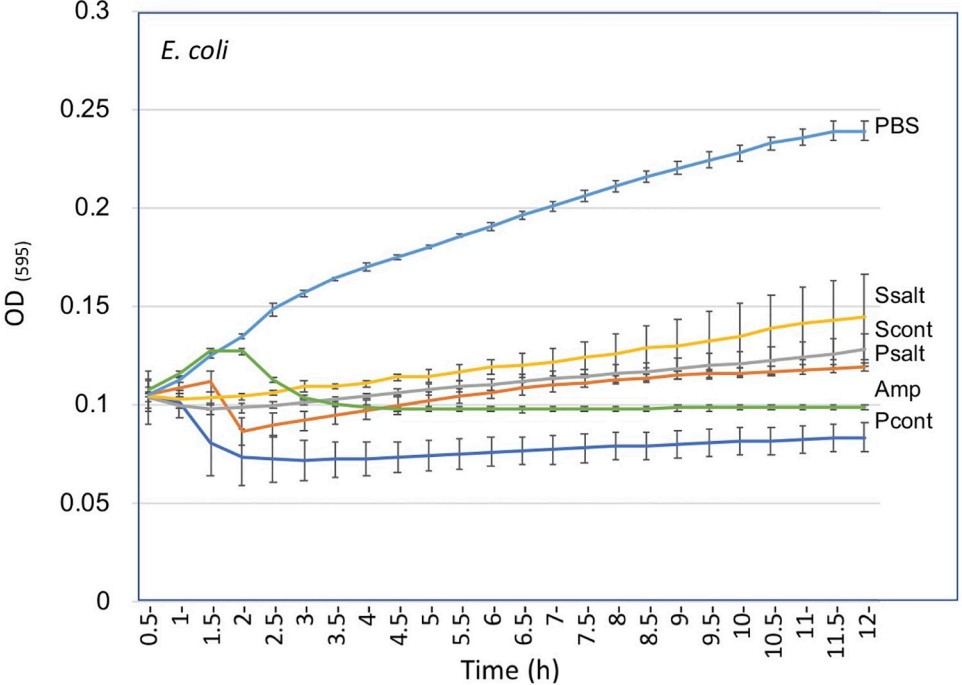

**Fig 7. Pericarps and seeds of *A. hierochuntica* release substances that inhibit bacterial growth.** *Escherichia coli* was grown in a flat-bottom 96-well microtiter plate in the presence of PBS, ampicillin (Amp, 50 µg/ml) or in the presence of substances released from seeds (Scont; Ssalt) and pericarps (Pcont; Psalt) of control and salt-treated plants. Bacterial growth was monitored by measuring the $OD_{595}$ of the culture at 30 min intervals in the course of 12 h. Each treatment was performed in triplicates and error bars represent the standard deviation.

## Discussion

The dead organs enclosing the embryo, which together constitute the dispersal unit (e.g., seeds, fruits) have long been thought to provide means for dispersal and a hard shell for protection of the embryo during storage in the soil. However, in recent years, the dead shells enclosing the embryo including the seed coat, the fruit coat (i.e., pericarp) and floral bracts in grasses appear to have an elaborated function characterized by their inherent capacity of storage hundreds of proteins that can persist in active form for decades [11]. In addition to proteins other substances are stored within DOEEs providing hormones and metabolites to assist in controlling embryo development and germination.

Here, using the desert annual plant *Anastatica hierochuntica* we showed that the composition and level of proteins and other substances are affected by the growth conditions to which parental plants exposed during flowering and seed production. The effect of parental plant growth conditions on the properties of progeny seeds has been well documented demonstrating major impacts on seed size, color, dormancy and germination [21]. In his review article, Fenner [19] surveyed the literature (field observations and controlled experiments) documenting the effect of parent environment on germinability and suggested that certain environmental conditions appear to have similar effect over a wide range of species. Accordingly, exposure of parental plants to high temperatures, short days, red light, drought and high nitrogen levels in the course of seed production often resulted in lower dormancy and higher germinability.

We demonstrated a negative effect of parental growth conditions on seed properties showing increased abortion following exposure of parental plants to salt and salt+heat (S+H). Furthermore, progeny seeds derived from stressed plants had low germination capacity (about 65%) and reduced germination rate (2-fold reduction) compared to seeds derived from control, untreated plants. Similarly, seeds of the facultative halophyte *Anabasis setifera* and of the grass *Aeluropus lagopoides* collected from non-saline and saline habitats showed significantly better seed germination of the non-saline seeds compared to seeds collected from saline habitat [46, 47]. Yet, while many reports addressed the effect of parental conditions on the morphology and biology of progeny seeds, there are few reports addressing the effect of parental growth conditions on the composition and level of substances accumulated within DOEEs. Accordingly, exposure of *Arabidopsis* plants to low temperatures during seed maturation resulted in increased dormancy due to alteration in seed coat properties, that is, increased concentration of procyanidins [48].

Proteome analysis revealed hundreds of proteins released from the dead pericarps of *A. hierochuntica* following hydration, many of which are hydrolases and oxireductases or involved in response to stimulus. Several protein groups are of particular interest because of their potential involvement in seed germination including ROS detoxifying enzymes such as superoxide dismutase, catalase and peroxidase [49,50] and cell wall modifying enzymes [51–53]. Exposure of parental plants to salt or S+H stress resulted in changes in the composition and level of proteins and substances released from pericarps and from seeds following hydration. Accordingly, we found 38 DP proteins released from pericarps derived from control and salt-treated plants. Functional categorization of DP proteins highlighted protein involved in response to stress. This includes the down regulated protein RESPONSIVE TO DEHYDRATION 19 (RD19), a cysteine proteinase induced by desiccation and is required for resistance against the pathogenic bacterium *Ralstonia solanacearum* [54]. In addition, upregulated proteins include the PLAT DOMAIN PROTEIN 1 (PLAT1) a member of the stress protein family involved in mediating response to biotic and abiotic stresses [41,55], Annexin D2 protein a calcium-binding protein that is upregulated in heat stress [56] as well as Polygalacturonase inhibitor 2 (PGIP2) induced by fungal infection and is implicated in resistance to fungal pathogen [57,58].

The analysis of proteins secreted from seeds of *A. hierochuntica* revealed only one DP protein, which is upregulated in progeny seeds derived from stress-treated plants, namely defensin, also known as antimicrobial peptides (AMPs) that are induced following pathogen attack [44]. Accordingly, the level of Defensin-like protein 195 (whose homolog in *Arabidopsis* is encoded by At2g43510) was increased 24 and 11.5-fold in seeds derived from salt and S+H plants, respectively, compared to control, untreated plants. Indeed, defensins were reported to be upregulated in various plant species subjected to abiotic stress conditions including salt, drought and cold stress [44,59,60]. It appears that exposure of mother plants to abiotic stresses primes progeny seeds to biotic stresses mediated by defensin, which might lead to improved resistance of germinating seeds to potential soil pathogen. Priming resistance to virulent bacteria has been reported in *Arabidopsis thaliana* following exposure to heat, cold and salt [61], mechanical stress [62] and wounding [63]. Also, submergence of *Arabidopsis* plants for 1 h conferred resistance to the virulent bacterial pathogen *Pseudomonas syringae* pv. tomato DC3000 (Pst DC3000), which is mediated by the transcription factor WRKY22 [64].

Apparently, exposure of mother plants to stress conditions in the course of flowering and seed production strongly affect progeny seed properties as well as the accumulation of substances in the pericarp of *A. hierochuntica*. This is demonstrated by the reduction in nuclease activity in pericarps of stress-treated plants and increase protease activity in pericarps of S+H-treated plants. Moreover, a significant reduction in precursors of the TCA cycle (i.e., citrate, succinic acid, malic acid and fumaric acid) was observed in dead pericarps of salt-treated plants. We assume that the status of these TCA intermediates in dead pericarps might reflect their abundance in the live pericarp tissues in response to salt. Generally, plants subjected to saline soils undergo physiological stress that impacts essential cellular processes such as respiration and photosynthesis. The role played by mitochondria (where respiration and energy production occur) in plant response to salt has been documented revealing that about 50% of the TCA cycle proteins decreased in abundance in salt-tolerant plants [65]. This observation suggests that similar effect is exerted on the essentially salt-tolerant plant *A. hierochuntica* whose exposure to salt (200 mM) resulted in reduction of TCA cycle proteins and consequently to reduction in synthesis of the TCA cycle intermediates (citrate, succinic acid, malic acid and fumaric acid).

We also observed changes in levels of phytohormones including IAA, ABA and salicylic acid in pericarps derived from stress-treated plants. ABA is considered as a stress hormone whose level is commonly increased under various stress conditions including salinity [66,67]. However, we observed that ABA is significantly reduced in pericarps following exposure of mother plants to salt stress. Notably, elevation in ABA levels is commonly reported for glycophytes (salt-sensitive plants) such as *Arabidopsis* and tobacco while in the halophytic plant *Suaeda maritima*, it was reported that ABA levels are reduced following exposure to salt [68]. Likewise, exposure of the mangrove *Avicennia marina* seedlings (grown in pots from propagules) to 60% and 90% of seawater resulted in physiological responses, which are not mediated by increase in ABA levels [69]. The desert plant *A. hierochuntica* as well as *Eutrema salsugineum* are extremophytes and essentially stress tolerant species, which appears to be primed for stress [70]. Thus, their responses to the kind of stresses commonly prevail in their desert ecosystem, differ significantly from the response of glycophytic species. Although ABA is well known for its effect on seed dormancy [71], we could not detect any effect on germination of *A. hierochuntica* seeds. We assume that ABA level in pericarps (~25 ng/gDW) is too low for inhibition of germination but might be sufficient to allow for ABA signaling to induce stress response [71,72].

The antimicrobial activity in pericarp extracts appears to be stronger that in seed secretion. Yet, the strength of inhibition was reduced in seed secretion and pericarp extract obtained from salt-treated plants. The effect of DOEEs on bacterial growth appears to be species specific.

Thus, while seeds and pericarps of *Anastatica* release upon hydration potent antimicrobial substances [10], seeds and pericarps of *S. alba* were shown to release microbial promoting substances [12], which might enhance seedling growth by secreting growth factors or other substances that inhibit growth of potential pathogens [73,74]. These unique properties of seeds and DOEEs probably reflect the specific habitat of each plant species and might represent co-evolution with their specific microbiota.

Interestingly, seeds and pericarps of *A. hierochuntica* release upon hydration allelopathic substances (germination inhibitors) that act in a species-specific manner to inhibit seed germination of certain plant species. Thus, seeds and pericarps possess substances that strongly inhibited germination of *S. alba* seeds, but had no or slight effect on seed germination of *B. juncea* or *A. hierochuntica*. The possession of selective allelopathic substances might be beneficial in several ways including prevention of germination of potentially competitor plants [75] or to allow for facilitative plant-plant interaction to occur [76].

In summary, the effect of mother plant growth conditions on progeny seed properties including seed dormancy and seed germination has been well documented [21]. Consistent with previous reports we found changes in properties of the *Anastatica* progeny seeds when mother plants were subjected to stress conditions during flowering and fruit maturation. Germination of seeds derived from stress plants is significantly reduced and production of aborted seeds is increased, particularly in plants exposed to heat stress. Seed abortion is most prominent in glycophytes, such as *Arabidopsis thaliana* showing almost 90% abortion following short time (12 h) exposure to 200 mM NaCl [77]. Exposure to heat of many plant species during the flowering phase disrupts seed production and reduced agricultural crop yield [78,79]. Some of the effects of mother plant growth conditions (day length, altitude, salinity) on progeny seed behavior may be mediated by changes in seed coat properties, e.g., thickness, color [21]. We showed that properties of the pericarps were altered as a result of mother plant growth conditions. We observed changes in proteins released from *Anastatica* pericarps and their enzymatic activities as well as the capacity to inhibit seed germination of heterologous plant species and microbial growth. Obviously, DOEEs have become a central component of plant reproduction and seed biology and ecology. We should take DOEEs into consideration when addressing various aspects related to seeds. This includes beneficial substances in DOEEs that can increase food and feed quality, seeds as a fundamental agricultural entity where DOEEs can improve germination rate and capacity and confer seedling vigor as well as seeds as the basic element for conservation of genetic resources in gene banks where DOEEs can increase viability and germination capacity under storage condition. Finally, we have to take into consideration that the effect exerted by climate change on plant biodiversity and population dynamics can be mediated, at least partly, through the effect on DOEE properties, which in turn impact seed performance (including persistence in the soil and germination) and fate, soil microbiota and soil fertility [80–82].

## Supporting information

**S1 File.**
(PDF)

**S2 File.**
(XLSX)

## Acknowledgments

We thank S. Lev-Yadun for critical reading of the manuscript.

## Author Contributions

**Conceptualization:** Yitzchak Gutterman, Zhenying Huang, Gideon Grafi.

**Formal analysis:** Janardan Khadka, Buzi Raviv, Bupur Swetha, Rohith Grandhi, Jeevan R. Singiri, Nurit Novoplansky, Ivan Galis, Gideon Grafi.

**Funding acquisition:** Ivan Galis, Gideon Grafi.

**Investigation:** Janardan Khadka, Buzi Raviv, Bupur Swetha, Rohith Grandhi, Jeevan R. Singiri, Nurit Novoplansky.

**Methodology:** Ivan Galis, Gideon Grafi.

**Project administration:** Nurit Novoplansky.

**Resources:** Yitzchak Gutterman.

**Supervision:** Gideon Grafi.

**Writing – original draft:** Gideon Grafi.

**Writing – review & editing:** Janardan Khadka, Yitzchak Gutterman, Ivan Galis, Zhenying Huang.

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
