## [Decision Letter · Decision Letter 0]

24 Jun 2020

PONE-D-20-13034

Maternal environment alters dead pericarp biochemical properties of the desert annual plant Anastatica hierochuntica L.

PLOS ONE

Dear Dr. Grafi,

Thank you for submitting your manuscript to PLOS ONE. After careful consideration, we feel that it has merit but does not fully meet PLOS ONE’s publication criteria as it currently stands. Therefore, we invite you to submit a revised version of the manuscript that addresses the points raised during the review process.

 The work presented in this manuscript has interesting results. Reviewers have pointed out that in the present form, it lacks methodological details, rationale for not having an important treatment, and details about statistical tools.  Authors can re-work on the suggestions to improve overall presentation of their manuscript.

We look forward to receiving your revised manuscript.

Kind regards,

Suprasanna Penna

Academic Editor

PLOS ONE

Journal Requirements:

2. In your Methods section, please provide additional information regarding the permits you obtained for the seed collection.

Please ensure you have included the full name of the authority that approved the field site access and, if no permits were required, a brief statement explaining why.

Additional Editor Comments:

The work presented in this manuscript has interesting results. Reviewers have pointed out that in the present form, it lacks methodological details, rationale for not having an important treatment, and details about statistical tools.  Authors can re-work on the suggestions to improve overall presentation of their manuscript.

Reviewers' comments:

Reviewer's Responses to Questions

**Comments to the Author**

1. Is the manuscript technically sound, and do the data support the conclusions?

Reviewer #1: Yes

Reviewer #2: Partly

2. Has the statistical analysis been performed appropriately and rigorously? 

Reviewer #1: No

Reviewer #2: I Don't Know

3. Have the authors made all data underlying the findings in their manuscript fully available?

Reviewer #1: Yes

Reviewer #2: Yes

4. Is the manuscript presented in an intelligible fashion and written in standard English?

Reviewer #1: Yes

Reviewer #2: Yes

5. Review Comments to the Author

Reviewer #1: In the proposed manuscript, the authors around G. Grafi have pursued their longstanding research topic to understand the role of dead maternal seed tissues on embryo performance during germination and the impact of environmental factors on the process. The manuscript represents a solid scientific work which based on usage of numerous experimental methods to draw the conclusions and support the hypothesis.

Despite of this, there are some points which have to be improved before the manuscript can be accepted for publication.

pp. 5-6 (material and methods), p. 10 and Fig. 2: The major question is concerning the germination assay. The authors should provide how many seeds and replications they used to show inhibitory effect of seed exudates and pericarp extracts on germination of the seeds under study. Similarly, it is not enough to show the inhibitory effect by pictures. Quantification of germination with appropriate statistic processing will have more power of evidence and improve the manuscript.

p. 9 and Fig. 1: the authors observed that stressed plants produced equal numbers of pods and seeds per plants as controls but had more aborted seeds. How can it be? Did stressed plants produce more flowers? The authors should clarify this phenomenon.

Fig. 1D and E: statistic analysis should be applied to compare germination of stressed seeds in relation to the control seeds.

Small remarks and mistakes:

P. 6: different font sizes in first and second paragraphs in “Proteome analysis” chapter.

P. 13: description of hormones: please use full names with abbreviations in brackets only at first mentions and only abbreviations further in the text.

Fig. 7: please use real time (e.g., in hours or minutes) instead numbering for X-axis. This will improve understanding of the figure.

After improving, the manuscript can be published in the Journal.

Reviewer #2: The manuscript deals with an interesting and important topic in plant reproductive biology with implications for plant growth and development. This is an important and comprehensive work that should eventually be published and will add significantly to the literature. I am a bit concerned by the lack of details in a few areas of methods, and lack of rationale for not having an important treatment, and lack of description of statistics. Kindly address the concerns and resubmit.

Line 99-102: They provided the overview of results in the last lines of introduction, while this is common now a days, it would be better to discuss the hypotheses here.

Line 107: too vague, where in the desert, what kind of soil, conditions while collecting the seeds?

Line 108: What is standard gardening soil?

Line 114: Why was the treatment (Heat) by itself excluded?

Line 121: This is a bit confusing? Are these self-pollinating flowers? What is the seed set rate and variation among the plants? Without describing the mating system of the species, it is really hard to understand these treatment effects.

Line 128 -132: The authors really need to understand what materials and methods mean. This part is designed to provide enough details so others can replicate. No details on how the extraction was done (lyophilizer? Mortar and Pestle, under liq N2?) or details on centrifuge and shaking are provided.

Line 136: Sample size? How many seeds per petri dish? When was the germination rate data collected?

Line 161; from 4 different plants?

A paragraph on statistical analyses and their details needs to be added. It is unlikely that the raw data is normally distributed. So explain what kind of analyses were done and their details.

Page 15 and 16: Break it into 2-3 paragraphs. It is extremely tedious to read this, all though everything is relevant and well written.

6. PLOS authors have the option to publish the peer review history of their article (what does this mean?). If published, this will include your full peer review and any attached files.

Reviewer #1: No

Reviewer #2: No

---

## [Author Response · Author response to Decision Letter 0]

14 Jul 2020

Dear Editor,

Thanks for the opportunity you granted us to revised our manuscript entitled “Maternal environment alters dead pericarp biochemical properties of the desert annual plant Anastatica hierochuntica L. by Khadka et al.

We would like to thank the Reviewers for their thoughtful comments. These comments were instrumental in revising and amending our manuscript. Thus the manuscript was modified in light of Reviewers’ comments, which we addressed point by point below.

We would like to state that:

1. The manuscript meets PLOS ONE's style requirements

2. Information regarding seed collection was added in M&M.

3. The original uncropped and unadjusted images underlying gel results presented in Fig 4 were included in supplementary information (S6 Fig).

4. All data regarding this study are included in the manuscript and supporting information.

Response to Reviewers

Reviewer #1: In the proposed manuscript, the authors around G. Grafi have pursued their longstanding research topic to understand the role of dead maternal seed tissues on embryo performance during germination and the impact of environmental factors on the process. The manuscript represents a solid scientific work which based on usage of numerous experimental methods to draw the conclusions and support the hypothesis.

Despite of this, there are some points which have to be improved before the manuscript can be accepted for publication.

Comment 1: pp. 5-6 (material and methods), p. 10 and Fig. 2: The major question is concerning the germination assay. The authors should provide how many seeds and replications they used to show inhibitory effect of seed exudates and pericarp extracts on germination of the seeds under study. Similarly, it is not enough to show the inhibitory effect by pictures. Quantification of germination with appropriate statistic processing will have more power of evidence and improve the manuscript.

Response to comment 1: Details about the germination assays are provided including number of replicates and number of seeds in each replicate (see M&M lines 143-144, and legend to Fig 2). We run more experiments and provided quantification and statistical analysis for the effect of pericarp extracts on germination of B. juncea and S. alba (new Fig 2)

Comment 2: p. 9 and Fig. 1: the authors observed that stressed plants produced equal numbers of pods and seeds per plants as controls but had more aborted seeds. How can it be? Did stressed plants produce more flowers? The authors should clarify this phenomenon.

Response to comment 2: This data may be confusing, because the data of Fig 1B (Average number of seeds per plant) refers to all seeds produced by a given plant, both normal and aborted seeds (this is noted in the legend to Fig 1).

Comment 3: Fig. 1D and E: statistic analysis should be applied to compare germination of stressed seeds in relation to the control seeds..

Response to comment 3: Statistical analysis was added (see Figure 1D and E).

Small remarks and mistakes:

P. 6: different font sizes in first and second paragraphs in “Proteome analysis” chapter.

Font sizes corrected.

P. 13: description of hormones: please use full names with abbreviations in brackets only at first mentions and only abbreviations further in the text.

Corrected.

Fig. 7: please use real time (e.g., in hours or minutes) instead numbering for X-axis. This will improve understanding of the figure.

Real time is now provided for the X axis of Fig. 7.

After improving, the manuscript can be published in the Journal.

Reviewer #2: The manuscript deals with an interesting and important topic in plant reproductive biology with implications for plant growth and development. This is an important and comprehensive work that should eventually be published and will add significantly to the literature. I am a bit concerned by the lack of details in a few areas of methods, and lack of rationale for not having an important treatment, and lack of description of statistics. Kindly address the concerns and resubmit.

Comment 1: Line 99-102: They provided the overview of results in the last lines of introduction, while this is common now a days, it would be better to discuss the hypotheses here.

Response to comment 1: Hypothesis and the rationale of our study is provided as suggested by the Reviewer (lines 78-82).

Comment 2: Line 107: too vague, where in the desert, what kind of soil, conditions while collecting the seeds?

Response to comment 2: Details regarding the collection site is provided in M&M (lines 102-105).

Comment 3: Line 108: What is standard gardening soil?

Response to comment 3: This is a potting soil mixture commonly used in our Institute for growing plants. This soil contains Sphagnum peat moss and perlite (2:1 ratio) supplemented with slow-release fertilizer (5 g/liter) (see lines 105-106).

Comment 4: Line 114: Why was the treatment (Heat) by itself excluded?

Response to comment 4: We initially attempted to examine certain abiotic stress conditions that prevail in the desert ecosystem during the growing season (winter time) from which seeds were collected. The desert ecosystem is characterized by salty soil, which represents a fixed and continuous stress condition, while heat stress may occur occasionally (usually for a few days) during the growing season (winter time). Thus, we assumed that it would be more relevant to address the effect of limited heat stress when given in combination with salt stress, rather than heat stress alone.

Comment 5: Line 121: This is a bit confusing? Are these self-pollinating flowers? What is the seed set rate and variation among the plants? Without describing the mating system of the species, it is really hard to understand these treatment effects.

Response to comment 5: Anastatica hierochuntica is a self-pollinating plant. We provided details on its mode of flowering in the text (lines 85-88). 

Comment 6: Line 128 -132: The authors really need to understand what materials and methods mean. This part is designed to provide enough details so others can replicate. No details on how the extraction was done (lyophilizer? Mortar and Pestle, under liq N2?) or details on centrifuge and shaking are provided.

Response to comment 6: We added missing details to the extraction procedure (lines 128-130).

Comment 7: Line 136: Sample size? How many seeds per petri dish? When was the germination rate data collected?

Response to comment 7: Details on sample size and rate data were added (see legend to Fig. 1 and lines 137-145).

Comment 8: Line 161; from 4 different plants?

Response to comment 8: Each replicate contains randomly selected seeds or pericarps from a pool of seeds/pericarps of 14-18 plants (see lines 167-169).

Comment 9: A paragraph on statistical analyses and their details needs to be added. It is unlikely that the raw data is normally distributed. So explain what kind of analyses were done and their details.

Response to comment 9: A paragraph on statistical analyses was added (see lines 265-269).

Comment 10: Page 15 and 16: Break it into 2-3 paragraphs. It is extremely tedious to read this, all though everything is relevant and well written.

Response to comment 10: Page 15-16 was separated into 3 paragraphs as suggested.

---

## [Decision Letter · Decision Letter 1]

20 Jul 2020

Maternal environment alters dead pericarp biochemical properties of the desert annual plant Anastatica hierochuntica L.

PONE-D-20-13034R1

Dear Dr. Grafi,

We’re pleased to inform you that your manuscript has been judged scientifically suitable for publication and will be formally accepted for publication once it meets all outstanding technical requirements.

Kind regards,

Suprasanna Penna

Academic Editor

PLOS ONE

Additional Editor Comments (optional):

All the suggestions and comments made by the reviewers have been suitably addressed in the revised version.

**Comments to the Author**

Reviewer: All comments have been addressed

2. Is the manuscript technically sound, and do the data support the conclusions?

Reviewer #2: Yes

3. Has the statistical analysis been performed appropriately and rigorously? 

Reviewer #2: Yes

4. Have the authors made all data underlying the findings in their manuscript fully available?

Reviewer #2: Yes

5. Is the manuscript presented in an intelligible fashion and written in standard English?

Reviewer #2: Yes

6. Review Comments to the Author

Reviewer #2: (No Response)

7. PLOS authors have the option to publish the peer review history of their article (what does this mean?). If published, this will include your full peer review and any attached files.

Reviewer #2: No

---

## [Editor Report · Acceptance letter]

22 Jul 2020

PONE-D-20-13034R1 

Maternal environment alters dead pericarp biochemical properties of the desert annual plant Anastatica hierochuntica L. 

Dear Dr. Grafi:

I'm pleased to inform you that your manuscript has been deemed suitable for publication in PLOS ONE. Congratulations! Your manuscript is now with our production department. 

Kind regards, 

on behalf of

Dr. Suprasanna Penna 

Academic Editor

PLOS ONE